# Spatial–Temporal Evolution and Driving Factors of Economic Dual Circulation Coordinated Development in China's Coastal Provinces

**Jingyi Wang [1], Shuguang Liu [1,2,*] and Yubin Zhao [1]**

[1] School of Economics, Ocean University of China, Qingdao 266100, China; wangjingyi5299@163.com (J.W.)
[2] Institute of Marine Development, Ocean University of China, Qingdao 266100, China
* Correspondence: 2000046@ouc.edu.cn

**Abstract:** China's coastal area is an important node, carrying the connection between internal and external circulation. It is of great significance to explore the spatial and temporal evolution of economic dual circulation coordinated development and its driving factors. In this paper, the Technique for Order Preference by Similarity to Ideal Solution (TOPSIS) evaluation model based on the Criteria Importance Though Intercriteria Correlation and entropy (CRITIC-entropy) weight method, coupling coordination model, standard deviation ellipse and exploratory spatial data analysis were used to analyze the spatial and temporal evolution characteristics of economic dual circulation coordinated development of China's coastal area. In addition, the geographical detector was employed to identify its driving factors. The results showed that: (1) The development level of internal and external economic circulation in China's coastal area was mainly stable and rising in a fluctuating manner, and the level of internal circulation was higher than that of external circulation. The overall coupling coordination degree of economic dual circulation exhibited a positive trend. (2) There was regional heterogeneity and spatial correlation in the coupling coordination degree of economic dual circulation in the coastal area of China, and the spatial distribution pattern showed the characteristic of being "strong in the internal and weak in the external". (3) The coordinated development of economic dual circulation was driven by multiple factors. Regional technological innovation capability, per capita income level, circulation development level, marketization process, digitization level and financial development level were the core driving forces. Based on the findings of this paper, a series of policy recommendations for improving the coordination development between internal and external economic circulation was proposed.

**Keywords:** economic dual circulation; coupling coordination degree; spatial–temporal evolution; driving factor; China's coastal area

## 1. Introduction

Since the reform and opening up, China has implemented an export-oriented international circulation economy development strategy and integrated into the global economic system with a comparative advantage in terms of relatively cheap labor costs, leading to the rapid economic growth and urbanization and industrialization development [1,2]. From 1978 to 2016, the average annual growth rate of China's GDP was 9.7%, far exceeding that of other major economic powers in the same period. However, excessive dependence on international circulation led to huge risks in the economic development. Problems such as the "low value locking" in the industrial chain, containment of key areas and core technologies, regional development disequilibrium, insufficient economic development impetus and structural imbalance arose [3,4]. In recent years, with the prevalence of anti-globalization trends represented by unilateralism and trade protectionism and rising geopolitical risks, superimposed on the Black Swan and Gray Rhino events, global turbulence causes and

risk points increase and uncertainty and instability of the external environment rises significantly, bringing new difficulties and challenges to China's economic development [5]. General Secretary Xi Jinping stressed the need to accelerate the establishment of a new pattern of development in 2022, which was listed in the 14th Five-Year Plan. As an economy with the huge domestic market advantage and domestic demand potential, smooth operation and coordinated development of internal and external economic circulation becomes the key to enhancing China's endogenous power and mitigating external risks.

Several scholars have researched economic dual circulation but mainly focus on the conceptual and theoretical analysis and the literature lacks qualitative research on the driving factors of economic dual circulation. More attention is paid to national differences but there is no research on a regional scale for analysis. To fill the gap, this paper looked at the coastal provinces in China and attempted to study the spatial and temporal characteristics and driving factors of the coordinated development of economic dual circulation through an empirical analysis. The possible marginal contributions of this paper are as follows: Firstly, the commodity circulation, element flow and income distribution were included in the circulation link to construct the evaluation index system of the development level of economic internal and external circulation, which makes the measurement results more scientific and reliable. Secondly, the historical evolution law and spatial distribution characteristics of coupling coordination degree of economic dual circulation in the coastal area were explored from hierarchical distribution, spatial distribution pattern and spatial agglomeration degree, and the driving factors of the coordinated development of economic internal and external circulation were further tested. Thirdly, China's coastal area is a vital node, carrying the connection between the internal circulation and external circulation. This paper took China's coastal area as the study area and explored its economic dual circulation; this can provide a reference for the coastal area and even the whole country to retain a competitive edge and achieve high-quality development in the complex and changing external economic environment.

The rest of this paper is arranged as follows. Related materials are introduced in Section 2, after which evaluation index systems are constructed and research methods are provided in Section 3. Section 4 analyzes the spatial and temporal evolution characteristics of economic dual circulation and tests its driving factors and their interactions. Section 5 is a discussion. The conclusions, implications and policy recommendations are presented in Section 6.

## 2. Literature Review

### 2.1. The Definition and Measurement of Economic Dual Circulation

The concept of economic circulation can be traced back to Francois Quesnay's circulation thought in the economic table. In his book *economic table*, he analyzed the distribution and reproduction of aggregate social products in French at that time [6], creating a precedent for the study of economic circulation theory. In the 1860s, Marx (2004) put forward the theory of social reproduction, which incorporated production, distribution, circulation and consumption into the process of capital circulation and divided the process into three sub-processes including monetary capital, production capital and commodity capital [7]. At the end of the 20th century, Leontief (1991) constructed a set of input–output tables and the input–output model and pointed out that production networks in the economic system often have complex structural relationships that are interrelated [8], which contributes to the emergence and development of the concept and analytical method of economic circulation. Although scholars have not yet reached a consensus on the definition of economic circulation, they tend to agree that economic circulation is the flow cycle process of elements and products among different subjects. The economic internal circulation occurs at home and the economic external circulation partly occurs abroad [4,9].

In terms of the measurement research of economic circulation, Grassman (1980) used the foreign trade dependence ratio to estimate the degree of dependence of a country's economy on other countries' economies [10]. However, there was a repeated statistical

problem with this indicator [11–14] and it did not consider the large share of intermediate trade [15]. In view of the limitations of this indicator, some scholars have proposed the employment of value-added trade volume to replace it [16–18]; for instance, Koopman et al. (2014) established a unified framework for value-added trade accounting [19]. On this basis, many scholars began to decompose the added value, distinguishing and measuring the internal circulation and external circulation. Chen et al. (2022) estimated the relative degree of China's domestic and international circulation by the proportion of China's added value depending on domestic final demand and foreign final demand based on the input–output model [20], and Lu et al. (2022) used the Asian Development Bank (ADB) input–output table to analyze the proportion structure of China's internal and external circulation from 2008 to 2020 [21]. Some scholars, such as Wang et al. (2022), constructed an index system to evaluate the development level of internal and external economic circulation [22].

## 2.2. The Relationship between Economic Internal Circulation and Economic External Circulation

During the 19th century, economic growth theory and international trade theory were successively put forward, emphasizing the crucial role of domestic demand and foreign demand in economic growth, as well as the positive interaction between them [23]. The relationship between internal circulation and external circulation has begun to receive widespread attention in academia. On the one hand, economic internal circulation can promote the development of economic external circulation. First, the increase in domestic demand will bring about an increase in domestic production capacity, expanding the scale of import and export and overseas markets [24]. Second, the internal circulation can improve the transfer efficiency of elements and products, accelerate resource integration and achieve industrial and technological upgrading, enhancing the competitiveness in global value chains [25,26]. On the other hand, economic external circulation will also promote the development of economic internal circulation through the resource allocation effect, technology diffusion effect, product competition effect and market size effect [27]. First, it can alleviate domestic employment pressure, increase residents' income and then expand domestic demand and stimulate investment [28]. Second, product imports will introduce new technologies and bring technology spillovers, increasing the domestic supply capacity and promoting the upgrading of the domestic industrial structure [29]. Third, the external circulation may intensify product competition and force domestic enterprises to improve product quality and competitiveness [30,31]. Fourth, under the condition of economies of scale, external circulation can reduce production costs and improve production efficiency, thus improving domestic market performance [32–34]. Therefore, scholars generally believe that the internal and external circulation are not separated from each other but interrelated and mutually reinforcing [35]. However, each economy should lay a different emphasis at different stages of economic development [36].

## 2.3. The Driving Factors of Economic Dual Circulation

Scholars have analyzed the driving factors of economic dual circulation from different perspectives and found that industrial structure upgrading [37], digital economy development [38,39], circulation system construction and perfection [40], financial technology development [41], marketization development [42] and so forth are important factors to promote the economic dual circulation development. Among them, the upgrading of the industrial structure can increase the added value of products [43] and optimize the supply structure and consumption structure [44] internally and link the global value chain to obtain external advanced knowledge [45] externally, bringing about the development of economic dual circulation. Digital development contributes to dual circulation development by reducing external costs [46,47], improving total factor productivity [48] and enhancing the economic system resilience. As a crucial means of unimpeded dual circulation [40], an efficient circulation system can achieve the mutual promotion of supply and demand, cost reduction, efficient allocation and efficiency improvement. Technological innovation is the crux of breaking the blockade on techniques and enhancing domestic productivity and

international competitiveness [49,50]. The level of financial development and the level of marketization help to speed up the smooth flow and rational allocation of elements and stimulate the vitality of economic development, thereby promoting the coordinated development of dual circulation [42]. In addition, the role of the economic foundation cannot be ignored [51].

## 3. Materials and Methods

### 3.1. Study Area

The eastern coastal area is the frontier zone of China's economic development and plays a vital role in promoting the development of China's economic dual circulation. By the end of 2021, the total population of China's coastal area was 637 million and the regional GDP was CNY 60.43 trillion, accounting for 45.08% and 53.06% of that of the whole country, respectively. This paper selected 11 coastal provinces (municipalities or autonomous regions) in China as the research area (Figure 1), namely Liaoning, Hebei, Tianjin, Shandong, Jiangsu, Shanghai, Zhejiang, Fujian, Guangdong, Guangxi and Hainan from north to south.

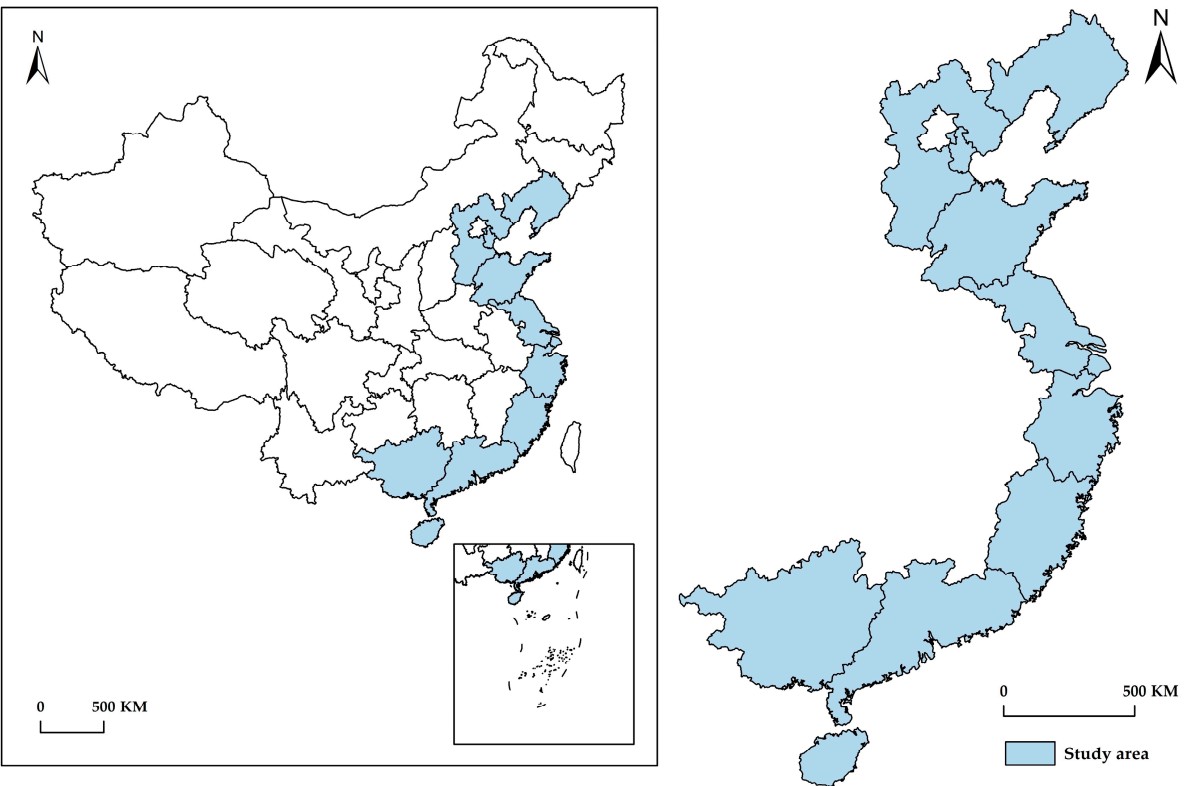

**Figure 1.** Study area. (Remark: based on the standard map GS (2019) No. 4345 of the standard map service website of the Ministry of Natural Resources; the base map boundary was not modified).

### 3.2. Construction of the Indicator System and Data Sources

The economic external circulation mainly includes the circulation process of products and services, capital and technology between the domestic and international markets [9]. Drawing on the selection of relevant indicators in the existing literature [51,52], this paper established an evaluation index system of economic external circulation (Table 1). In terms of capital openness, FDI and the flow of non-financial OFDI was chosen to represent the introduction of foreign capital and investment abroad, respectively. In terms of trade openness, the total import and export volume and high-tech import and export volume were selected for characterization, in which the high-tech import and export volume was used to measure the upgrading of foreign trade.

**Table 1.** Evaluation index system of the development of economic internal and external circulation in the coastal area.

| First-Level Indicator | Second-Level Indicator | Third-Level Indicator | Specific Indicator (Character) |
|---|---|---|---|
| Economic external circulation | Capital openness | Foreign capital introduction | FDI (USD 10,000) (+) |
| | | Investment abroad | Flow of non-financial OFDI (USD 10,000) (+) |
| | Trade openness | Trade abroad | Total import and export volume (USD 100,000,000) (+) |
| | | Foreign trade upgrading | High-tech import and export volume (USD 1,000,000) (+) |
| Economic internal circulation | Production link | Production scale | Per capita fixed assets investment (10,000 CNY/person) (+) |
| | | Production efficiency | Labor productivity (%) (+) |
| | Circulation link | Commodity circulation | Proportion of total retail sales of consumer goods to regional GDP (%) (+) |
| | | Element flow | Net population change rate (‰) (+) |
| | | | Capital share growth rate (%) (+) |
| | | | Patent application and authorization growth rate (%) (+) |
| | Distribution link | Income distribution | Per capita income ratio of urban and rural residents (%) (−) |
| | Consumption link | Consumption level | Proportion of per capita consumption expenditure of residents to GDP (%) (+) |
| | | Consumption structure | Engel coefficient (%) (−) |

Relevant scholars have explored the evaluation index system of economic internal circulation from different perspectives. Zhao (2022) evaluated the economic internal circulation from consumption and production [52]. Wang and Sun (2023) set up an evaluation index system of internal circulation development level including 33 indicators in six aspects: the population urbanization, economic development, digital economy development, supply-side reform, social security system and public service system [53]. Combining the existing research [22] and Marx's theory of social reproduction, this paper constructed an evaluation system of economic internal circulation (Table 1) including 13 indicators from four aspects: production, distribution, circulation and consumption. The production link was the starting point and material basis of the circulation link and the per capita fixed assets investment and labor productivity were selected to characterize the production scale and production efficiency. The circulation link and distribution link directly affect the connection and smoothness of the production link and consumption link. It was described from three dimensions: commodity circulation, element flow and income distribution. The

proportion of total retail sales of consumer goods to regional GDP was chosen to measure commodity circulation. According to the practices of Lin and Wang (2006), Li and Chen (2007) and Keller and Yeaple (2009) [54–56], this paper chose the net population change rate, capital share growth rate and patent application and authorization growth rate to represent the flow of the three basic elements of labor, capital and technology. Specifically, the mechanical changes in population were calculated to represent the cross-regional mobility of the labor force [54] and the relative changes in the proportion of regional capital were used to measure the interprovincial mobility of capital [55]. The per capita income ratio of urban and rural residents was selected to represent income distribution. The consumption link is the end of social production and the starting point of a new round of reproduction, and the proportion of per capita consumption expenditure of residents to GDP and Engel coefficient were selected to characterize the consumption level and consumption structure.

Considering the accessibility of empirical data, this paper selected 2006–2020 as the research period. All the indicator data were from the China Statistical Yearbook (2007–2021) and Provincial Statistical Yearbooks and Statistical Bulletins (2007–2021). A few missing data were replaced by interpolation. For the variables affected by price fluctuations, such as regional GDP, total retail sales of consumer goods and per capita fixed assets investment, the price index using 2006 as the base period was used to deflate them.

### 3.3. Methods

#### 3.3.1. CRITIC-Entropy Weight Method

The CRITIC method determines the indices' objective weights according to the contrast intensity and conflict of evaluation indices [57]. The entropy weight method empowers each index by analyzing the amount of information reflected by the degree of variation of the index, that is, the greater the degree of variation, the more information the evaluation index can reflect and the higher the weight value. Taking into account the contrast intensity [58], conflict and variation degree among the indicators, this paper used the CRITIC-entropy weight method to comprehensively calculate the weight of the evaluation index of economic internal and external circulation development level in China's coastal area. The calculation steps are as follows:

Firstly, to eliminate the impact of different dimensions and units, the data of each original indicator are standardized, as shown in Formulas (1) and (2).

Positive indicator:

$$x_{ij} = \frac{X_{ij} - \min(X_{ij})}{\max(X_{ij}) - \min(X_{ij})} \tag{1}$$

Negative indicator:

$$x_{ij} = \frac{\max(X_{ij}) - X_{ij}}{\max(X_{ij}) - \min(X_{ij})} \tag{2}$$

where $i$ represents coastal provinces and $j$ represents evaluation indices, $X_{ij}$ and $x_{ij}$ are raw and processed data, respectively, and $\max(X_{ij})$ and $\min(X_{ij})$ are the maximum and minimum values, respectively.

Secondly, calculating the weight by the CRITIC method:

$$c_j = \sigma_j \sum_{k=1}^{n} \left(1 - r_{jk}\right) \tag{3}$$

$$w_{j1} = \frac{c_j}{\sum_{j=1}^{n} c_j}, j = 1, 2, \ldots n \tag{4}$$

where $c_j$ is the amount of information reflected by index $j$, $\sigma_j$ is the standard deviation of index $j$ and $r_{jk}$ is the correlation coefficient between index $j$ and index $k$.

Thirdly, calculating the weight by the entropy weight method:

$$p_{ij} = \frac{x_{ij}}{\sum_{i=1}^{m} x_{ij}} \tag{5}$$

$$e_j = -\frac{1}{\ln m} \sum_{i=1}^{m} p_{ij} \ln p_{ij} \tag{6}$$

$$w_{j2} = \frac{1 - e_j}{\sum_{j=1}^{n}(1 - e_j)} \tag{7}$$

where $p_{ij}$ is the probability of the occurrence of index $j$ of coastal province $i$ and $e_j$ is the information entropy of index $j$.

Fourthly, combining the CRITIC method and the entropy weight method to determine the weight of index $j$:

$$w_j = \beta w_{j1} + (1 - \beta) w_{j2} \tag{8}$$

This paper assumes that the two weighting methods are equally important, so $\beta = 0.5$ [59].

### 3.3.2. TOPSIS Evaluation Model

The TOPSIS evaluation model was first proposed by Hwang and Yoon (1981) [60]. It is a multi-objective decision-making method based on the relative proximity to the idealized target of the evaluation object [61,62]. This paper adopted the TOPSIS evaluation model to measure the development level of economic internal and external circulation of China's coastal area. The calculation formula is as follows:

Firstly, constructing a weighting matrix:

$$R = (y_{ij})_{m \times n} \tag{9}$$

$$y_{ij} = w_j x_{ij} \tag{10}$$

where $y_{ij}$ is the evaluation value of index $j$ of coastal province $i$.

Secondly, determining the positive ideal solution and negative ideal solution:

$$D_j^+ = (\max y_{i1}, \max y_{i2}, \cdots \max y_{in}) \tag{11}$$

$$D_j^- = (\min y_{i1}, \min y_{i2}, \cdots \min y_{in}) \tag{12}$$

Thirdly, calculating the Euclidean distance to the positive ideal solution and the negative ideal solution of coastal province $i$, respectively:

$$d_i^+ = \sqrt{\sum_{j=1}^{n} \left(D_j^+ - y_{ij}\right)^2} \tag{13}$$

$$d_i^- = \sqrt{\sum_{j=1}^{n} \left(D_j^- - y_{ij}\right)^2} \tag{14}$$

Fourthly, calculating the relative proximity to the idealized target of coastal province $i$:

$$U_{ij} = \frac{d_i^-}{d_i^+ + d_i^-} \tag{15}$$

### 3.3.3. Coupling Coordination Model

Based on the concept of capacity coupling in physics and the capacity coupling coefficient model [63], coupling measures the interaction between two (or more) systems

or elements. At the same time, considering the staggered, dynamic and unbalanced characteristics of different systems or elements [64], coordination judges the dynamic relationship of interdependence, coordination and mutual promotion under the benign interaction between systems or elements. This paper introduced the coupling coordination model to reflect the coupling coordination level of economic dual circulation. The formula of the coupling coordination model is as follows:

$$C = 2\sqrt{\frac{I \cdot E}{(I + E)^2}} \tag{16}$$

$$T = \alpha I + \beta E \tag{17}$$

$$D = \sqrt{C \cdot T} \tag{18}$$

where $C$ is the coupling degree, $I$ and $E$ are the development level of economic internal and external circulation, respectively, $T$ is the coordination degree and $\alpha$ and $\beta$ are undetermined coefficients. Since the new development pattern assigns a central role to the domestic market, the weight of $\alpha$ and $\beta$ were formulated as 0.7 and 0.3 according to the relevant research practices [52]; $D$ is the coupling coordination degree. Referring to the existing research [65], the coupling coordination degree of economic dual circulation is divided into six grades using the uniform distribution function method, and the section partition of the coupling coordination degree of each grade is shown in Table 2.

**Table 2.** Classifications of coupling coordination levels.

| Value Range | Class |
| --- | --- |
| 0.0–0.2 | Severe dissonance |
| 0.2–0.4 | Moderate dissonance |
| 0.4–0.5 | On the verge of dissonance |
| 0.5–0.6 | Primary coordination |
| 0.6–0.8 | Good coordination |
| 0.8–1.0 | High-quality coordination |

3.3.4. Standard Deviation Ellipse

Standard deviation ellipse is a method to reveal the overall characteristics of economic spatial distribution by the spatial distribution ellipse with the average center, major axis, minor axis and azimuth as the basic parameters [66]. The method was first proposed by Lefever (1926) [67] and has been widely used in economics, geography, sociology, ecology and other fields [68]. Specifically, the spatial distribution range of the ellipse and average center reflect the main area and relative position (center of gravity) of the overall spatial distribution of elements in two-dimensional space. The azimuth reflects the main trend direction of its distribution, and the major axis and minor axis represent the dispersion degree of elements in the primary and secondary trend directions. The main parameters of the standard deviation ellipse are calculated as follows:

Average center:

$$G(X, Y) = \left( \frac{\sum_{i=1}^{n} w_i x_i}{\sum_{i=1}^{n} w_i}, \frac{\sum_{i=1}^{n} w_i y_i}{\sum_{i=1}^{n} w_i} \right) \tag{19}$$

Azimuth:

$$\tan \theta = \frac{\left( \sum_{i=1}^{n} w_i^2 \widetilde{x}_i^2 - \sum_{i=1}^{n} w_i^2 \widetilde{y}_i^2 \right) + \sqrt{\sum_{i=1}^{n} w_i^2 \widetilde{x}_i^2 - \sum_{i=1}^{n} w_i^2 \widetilde{y}_i^2}}{2\sum_{i=1}^{n} w_i^2 \widetilde{x}_i \widetilde{y}_i} \tag{20}$$

Major axis standard deviation:

$$\sigma_x = \sqrt{\frac{\sum_{i=1}^{n} (w_i \widetilde{x}_i \cos\theta - w_i \widetilde{y}_i \sin\theta)^2}{\sum_{i=1}^{n} w_i^2}} \tag{21}$$

Minor axis standard deviation:

$$\sigma_y = \sqrt{\frac{\sum_{i=1}^{n} (w_i \widetilde{x}_i \sin\theta - w_i \widetilde{y}_i \cos\theta)^2}{\sum_{i=1}^{n} w_i^2}} \tag{22}$$

where $(x_i, y_i)$ refers to the spatial location of each coastal province, $w_i$ is the weight and $\widetilde{x}_i$ and $\widetilde{y}_i$ refer to the coordinate deviation from the location of each coastal province to the average center, respectively.

### 3.3.5. Exploratory Spatial Data Analysis

Exploratory spatial data analysis is a collection of a series of spatial analysis methods and techniques. It identifies spatial correlation and agglomeration and explains the spatial interaction mechanism between research objects by describing and visualizing spatial distribution patterns [69,70]. The core of exploratory spatial data analysis is the estimation of global spatial autocorrelation and local spatial autocorrelation.

Firstly, global spatial autocorrelation can evaluate whether the spatial distribution is clustering mode, discrete mode or random mode according to the location and attribute value of factors, which is mostly measured by Moran's I index [71]. Its formula is as follows:

$$I = \frac{n \sum_{i=1}^{n} \sum_{j=1}^{n} w_{ij} (U_i - \overline{U})(U_j - \overline{U})}{S^2 \sum_{i=1}^{n} \sum_{j=1}^{n} w_{ij}} \tag{23}$$

where $n$ is the sample size of the research area, $U_i$ and $U_j$ are the coupling coordination degree of economic dual circulation of coastal provinces $i$ and $j$ and $w_{ij}$ is the spatial weight matrix. Owing to the correlation between regional economy and geography, this paper selected the per capita real GDP gap reflecting the economic distance to construct the spatial weight matrix; $\overline{U}$ and $S^2$ are the mean and variance of the coupling coordination degree of economic dual circulation in the coastal area.

Secondly, local spatial autocorrelation further reveals the characteristics of local spatial interdependence and spatial heterogeneity [72]. This paper used Moran scatter plot to directly reflect the spatial agglomeration of the coordinated development of economic dual circulation in coastal provinces.

### 3.3.6. Geographical Detector

The geographical detector is a statistical method to detect spatial heterogeneity and reveal driving factors [73], including factor detection, interaction detection, risk area detection and ecological detection. Factor detection is mainly employed to test whether an influencing factor is the cause of the spatial differentiation of specific variables, and interaction detection can identify the interaction between different influencing factors. This paper adopted factor detection and interaction detection to investigate the driving factors of the coordinated development of the economic dual circulation in the coastal area. The model of factor detection is as follows:

$$q = 1 - \frac{\sum_{h=1}^{L} N_h \sigma_h^2}{N \sigma^2} \tag{24}$$

where $q$ represents the driving effect on the spatial differentiation of the coordinated development of economic dual circulation in the coastal area; its value range is [–1,1]. The closer it is to 1, the greater the driving effect of this factor. $L$ represents the number of categories of driving factors. Referring to the research of Zhu et al. (2015) and Wang et al. (2016) [74,75], this paper divided each driving factor into five grades using the natural

breakpoint method. $N_h$ and $N$ represent the sample number of the whole and secondary research area, respectively, and $\sigma_h^2$ and $\sigma^2$ are the corresponding variance of the coupling coordination degree of economic dual circulation, respectively.

The principle of interaction detection is to reveal the interaction between the different factors $X_i$ and $X_j$ on the spatial differentiation of the coordinated development of economic dual circulation of the coastal area and evaluate whether this interaction will enhance or weaken the driving effect of the single factor $X_i$ or $X_j$ by comparing $q(X_i)$, $q(X_j)$ and $q(X_i \cap X_j)$.

## 4. Results and Analysis

### 4.1. Spatial–Temporal Evolution Characteristics of Economic Dual Circulation Coordinated Development

4.1.1. Temporal Evolution Characteristics of Economic Dual Circulation Coordinated Development

As shown in Figure 2, the coupling coordination degree of economic dual circulation of the coastal area exhibited a fluctuating upward trend from 2006 to 2018, rising from 0.41 in 2006 to 0.59 in 2020 (an increase of 43.47%), and generally developing in the direction of benign coordination. The changes in the level of internal and external economic circulation development were mainly stable and rising in a fluctuating manner; the overall trends were consistent. However, the level of economic internal circulation development was obviously better than that of economic external circulation, indicating that the internal and external economic circulation of the coastal area have achieved the quality of development and the internal circulation development has great potential.

According to the change trend of the coupling coordination degree of economic dual circulation of the coastal area, it could be roughly divided into three stages: 2006–2011, 2012–2015 and 2016–2020. From 2006 to 2011, the coupling coordination degree of economic dual circulation of the coastal area was at a low level on the verge of dissonance. Among them, affected by the financial crisis in 2008, the external circulation fell into recession, leading to the stagnation of the dual circulation development of the coastal area. Since the reform and opening up, China's economic center of gravity has shifted from internal to external and the coastal area, as the main window of foreign trade, has achieved rapid economic development. However, the external circulation has long been embedded into the low end of the global value chain. Domestic demand had not been fully tapped, and the production structure was not optimized. The development quality of the economic dual circulation still had a lot of room for improvement. Therefore, China began to adjust the export-oriented economic development pattern and put forward the strategy of expanding domestic demand, laying a solid foundation for improving the development quality and coordinated development of internal and external economic circulation.

From 2012 to 2015, the coupling coordination degree of economic dual circulation in the coastal area displayed a relatively stable upward trend, from the verge of dissonance to primary coordination. The main reason was that during the Twelfth Five-Year Plan period, China further emphasized on building a long-term mechanism for expanding domestic demand, introduced timely supply-side structural reform and put forward clear guidance on economic restructuring, promoting the coordinated development of economic dual circulation.

From 2016 to 2020, the coupling coordination degree of economic dual circulation in the coastal area fluctuated widely and the growth rate declined. With the gradual increase of international uncertainties and unstable factors, the level of economic external circulation development in the coastal area fluctuated greatly and the economic development model that relied on the international market was more fragile. Moreover, the driving effect of the international market was weakening and the pulling effect of the domestic market was increasing. The development of economic dual circulation had entered a dynamic adjustment. At this stage, China further optimized and adjusted the development pattern, moderately expanded domestic demand and adhered to high-level opening to the outside

world, accelerating the improvement of the mutual influence and promotion of internal and external economic circulation and boosting high-quality economic development.

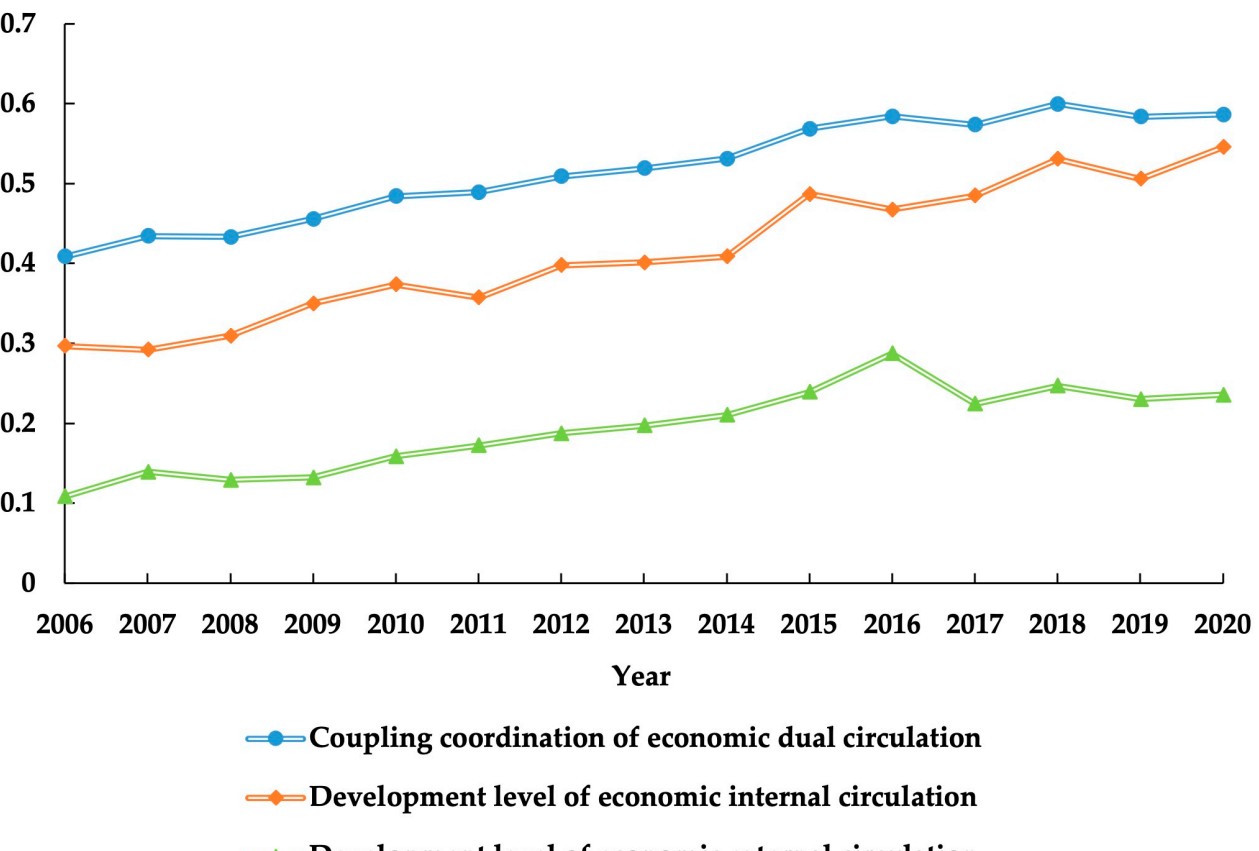

**Figure 2.** Temporal evolution of coupling coordination of economic dual circulation in the coastal area from 2006 to 2020.

4.1.2. Spatial Evolution Characteristics of Economic Dual Circulation Coordinated Development

From the perspective of the hierarchical distribution of the coupling coordination degree (Figure 3), there was obvious regional heterogeneity in the coupling and coordinated development of economic dual circulation in the coastal area from 2006 to 2020. Most provinces showed a good trend; however, some provinces also fell back. The coordination types were mainly primary coordination and good coordination, and there was no high-quality coordination. Among them, Shanghai, Jiangsu and Guangdong always maintained a high level and had been at the coordination level, indicating that these provinces, as the core area of China's foreign exchange and foreign capital and technology attraction, had become the pioneers of economic dual circulation development with their geographical advantages and strong economic foundation. The coupling coordination level of economic dual circulation in Zhejiang, Shandong, Tianjin, Fujian and Hebei were improved from the dissonance level to the coordination level, whereas Liaoning reached primary coordination in 2013 and then fell into the verge of dissonance. The possible reason was that Liaoning, whose brain drain was serious and economic growth rate was relatively slow, was in the throes of industrial supply-side structural transformation and upgrading. The improvement in economic dual circulation development still took time. Hainan and Guangxi transitioned from severe dissonance in 2006 to moderate dissonance in 2020, which was still at the level of dissonance, manifesting that, led by pioneers, the coordinated development of economic dual circulation in Guangxi and other regions significantly improved. However, due to weak innovation ability, a single industrial structure and weak economic power conversion,

Guangxi and other regions were still relatively backward in the process of economic dual circulation development.

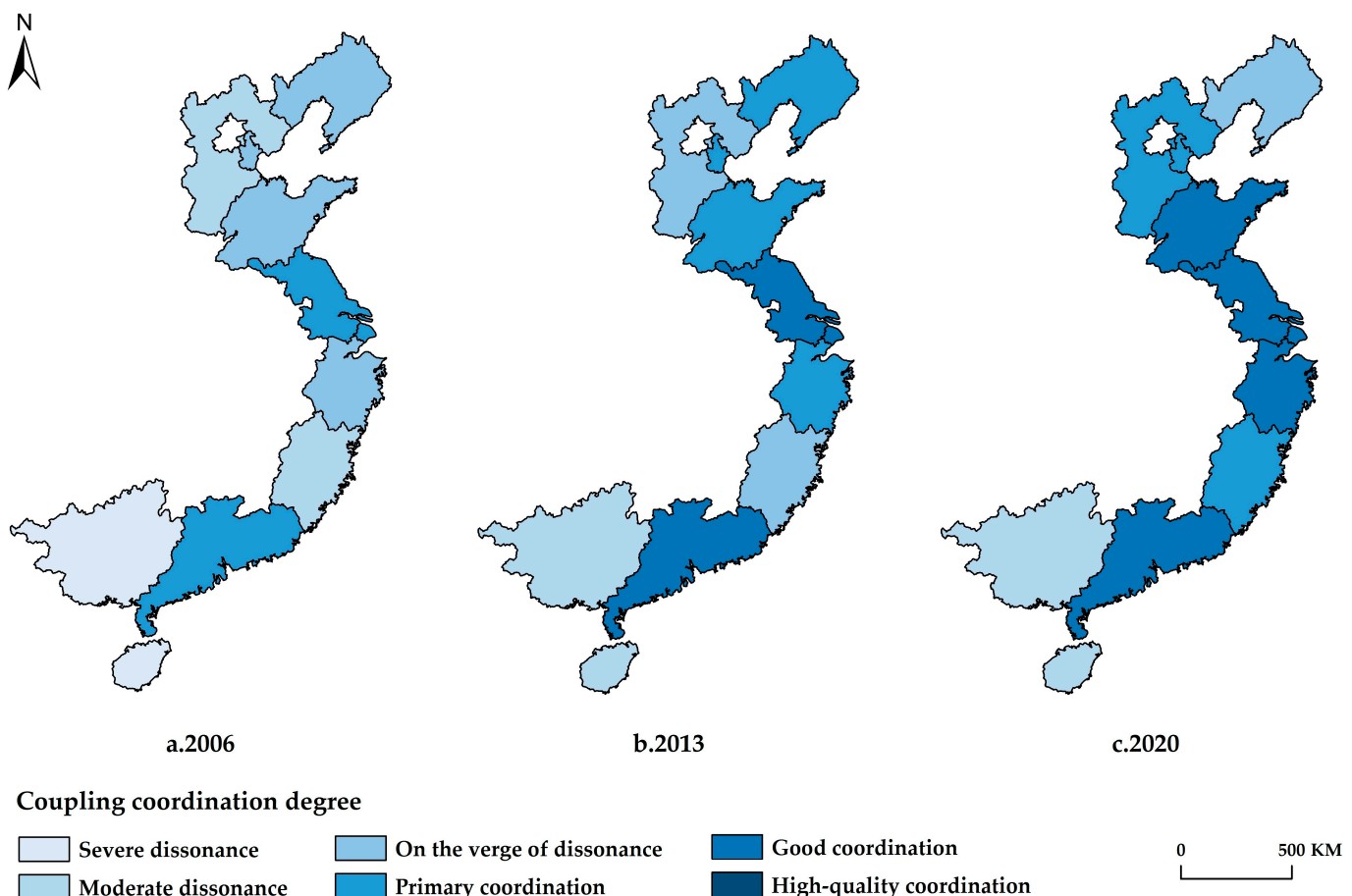

**Coupling coordination degree**

| | | |
|---|---|---|
| Severe dissonance | On the verge of dissonance | Good coordination |
| Moderate dissonance | Primary coordination | High-quality coordination |

**Figure 3.** Hierarchical distribution of coupling coordination of economic dual circulation in the coastal area in 2006, 2013 and 2020. (Remark: based on the standard map GS (2019) No. 4345 of the standard map service website of the Ministry of Natural Resources; the base map boundary was not modified).

From the perspective of the spatial distribution pattern of coupling coordination degree (Figure 4), the spatial distribution of economic dual circulation coupling coordination in the coastal area from 2006 to 2020 was dominated by the south (slightly west)–north (slightly east) direction, and the spatial distribution center was located in the middle and east. The coastal provinces of Guangdong, Zhejiang, Shanghai, Jiangsu and Shandong were distributed inside the standard deviation ellipse and the coastal provinces of Guangxi, Hainan and Hebei were distributed outside the standard deviation ellipse, showing the characteristic of "strong in the internal and weak in the external". Specifically, the center of gravity of the coordinated development of economic dual circulation in the coastal area was between 117.57° and 116.99° E and 31.28° and 31.79° N. From 2006 to 2013, it moved 24.15 km to the west by south, and from 2013 to 2020, it moved 53.03 km to the south by west, displaying that the improvement in the coupling coordination level of economic dual circulation in the southwest of the coastal area was significantly higher than that in the northeast. The expansion trend of the spatial distribution range reflected by the standard deviation ellipse was obvious. The standard deviation of the minor axis augmented from 389.79 km in 2006 to 410.12 km in 2020, which meant it was expanding. The major axis first extended and then shortened. It increased from 1051.49 km to 1101.65 km from 2006 to 2013, then decreased to 1073.31 km from 2013 to 2020. These results revealed that the spatial agglomeration degree of the coupling and coordination of economic dual circulation

in the coastal area reduced in the east–west direction but the distribution state in the north–south direction was unstable. The gap between the north and the south was still an important issue in the coordinated development of China's economy. The azimuth changed in a clockwise direction, from 10.9° in 2006 to 12.76°, displaying that with the construction of the Beibu Gulf Economic Zone and Hainan Free Trade Port, the pulling effect of provinces located in the southwest of the coastal area on the coupling coordination degree of the overall economic dual circulation of the coastal area enhanced and that the spatial pattern further evolved to the northeast–southwest direction. The northeastern provinces, such as Liaoning, needed to further stimulate the momentum and vitality of economic transformation and accelerate the coordinated development of economic dual circulation.

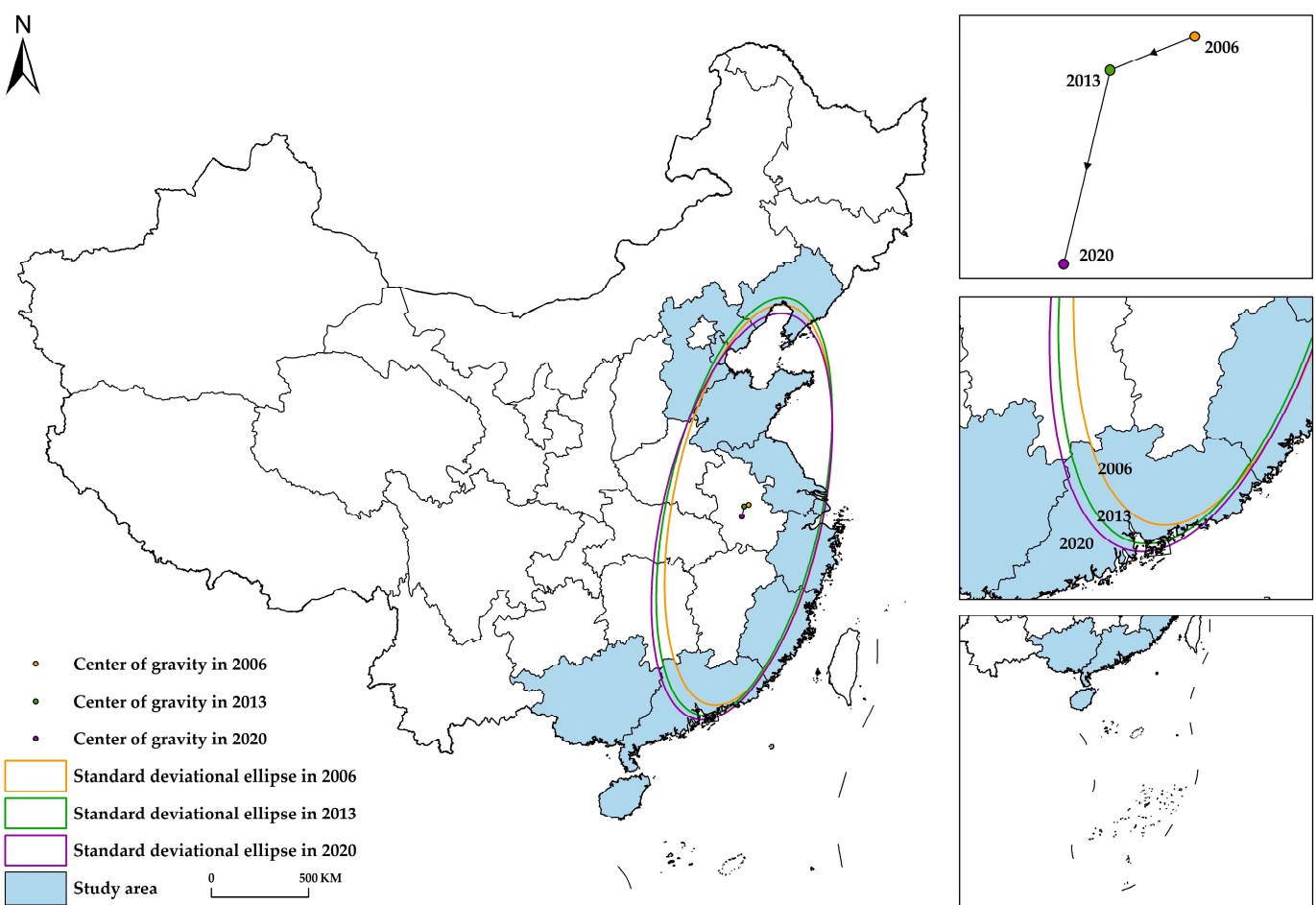

**Figure 4.** Spatial distribution pattern of coupling coordination of economic dual circulation in the coastal area in 2006, 2013 and 2020. (Remark: based on the standard map GS (2019) No. 4345 of the standard map service website of the Ministry of Natural Resources; the base map boundary was not modified).

From the perspective of the spatial agglomeration degree of the coupling coordination degree (Table 3), the Moran's I index of the coupling coordination level of economic dual circulation in China's coastal area from 2006 to 2020 was positive and passed the significance test. It revealed that there was a strong spatial positive correlation in the economic dual circulation of the coastal provinces. As shown in Figure 5, the coordinated development of economic dual circulation in the coastal area showed an obvious agglomeration trend in space, and the polarization characteristics were significant. During the research period, the spatial dependence of the coupling coordination level of economic dual circulation showed an inverted "U" trend, indicating that the agglomeration of the coordinated development

of economic dual circulation in the early stage was gradually strengthened. In recent years, the spatial positive correlation of adjacent provinces declined compared with previous years and the agglomeration situation slowed down.

**Table 3.** Global Moran's I index of coupling coordination of economic dual circulation in the coastal area from 2006 to 2020.

| Year | Moran's I | Z-Value |
|------|-----------|---------|
| 2006 | 0.306 *** | 2.373 |
| 2007 | 0.284 ** | 2.234 |
| 2008 | 0.308 *** | 2.377 |
| 2009 | 0.299 ** | 2.320 |
| 2010 | 0.314 *** | 2.424 |
| 2011 | 0.308 *** | 2.389 |
| 2012 | 0.334 *** | 2.543 |
| 2013 | 0.324 *** | 2.504 |
| 2014 | 0.341 *** | 2.581 |
| 2015 | 0.347 *** | 2.545 |
| 2016 | 0.347 *** | 2.560 |
| 2017 | 0.198 ** | 1.698 |
| 2018 | 0.212 ** | 1.759 |
| 2019 | 0.209 ** | 1.739 |
| 2020 | 0.181 * | 1.591 |

***, ** and * represent the significance levels of 1%, 5% and 10%, respectively.

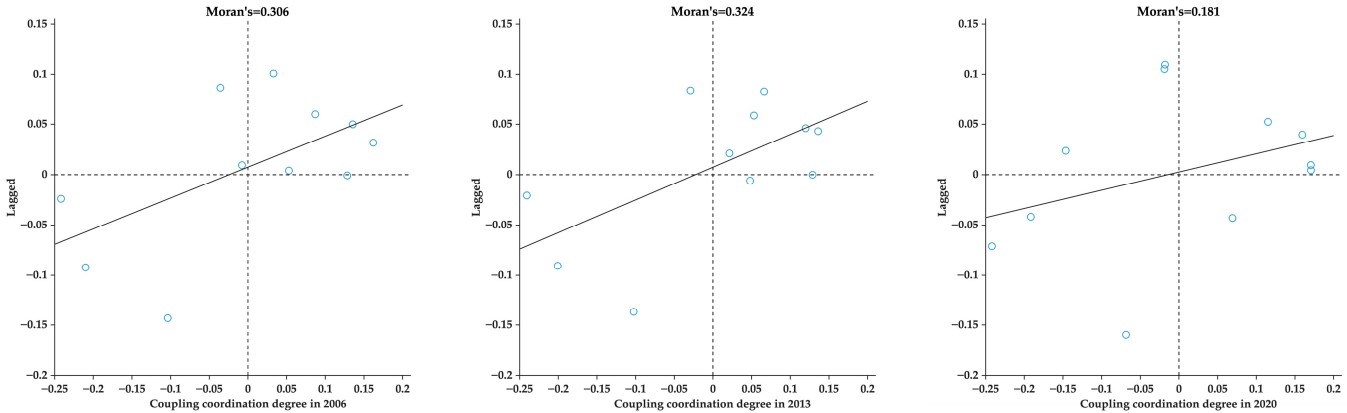

**Figure 5.** Moran scatter plot of coupling coordination of economic dual circulation in the coastal area in 2006, 2013 and 2020.

*4.2. Analysis of the Driving Factors of Economic Dual Circulation Coordinated Development*

4.2.1. Selection of Driving Factors

The coordinated development of internal and external economic circulation is the result of the interactions between a variety of factors. Therefore, combined with the existing research results [37–42,49–51] and driving mechanism analysis, this paper explored the coordinated development of economic dual circulation from the per capita income level (PGDP), industrial economic structure (INDU), financial development level (FINA), regional technological innovation capability (INNO), digitization level (DIGI), marketization process (MARK) and circulation development level (CIRC). Considering the representativeness and availability of data, the per capita GDP, proportion of the tertiary industry to GDP, proportion of the balance of deposits and loans of financial institutions to GDP, proportion of R&D expenditure to GDP, per capita volume of post and telecommunications business, marketization index and per area output value of circulation industry were selected for characterization.

### 4.2.2. Detection Results

From the view of individual effects (Table 4), different factors had different driving effects on the coordinated development of economic dual circulation in different years. Among them, the regional technological innovation capability (0.9571), per capita income level (0.9498), circulation development level (0.8852), marketization process (0.8374), digitization level (0.8008) and financial development level (0.6658) played a major role in the coordinated development of economic dual circulation, whereas the industrial economic structure (0.4216) played a much smaller role.

**Table 4.** Driving effect of factors on coupling coordination of economic dual circulation in 2006, 2013 and 2020.

| Factor | 2006 | 2013 | 2020 | Average |
|---|---|---|---|---|
| Per capita income level (PGDP) | 0.9691 *** | 0.9657 *** | 0.9145 *** | 0.9498 |
| Industrial economic structure (INDU) | 0.6149 *** | 0.3582 *** | 0.2917 *** | 0.4216 |
| Regional technological innovation capability (INNO) | 0.9607 *** | 0.9454 *** | 0.9653 *** | 0.9571 |
| Digitization level (DIGI) | 0.7718 *** | 0.8010 *** | 0.8295 *** | 0.8008 |
| Circulation development level (CIRC) | 0.9467 *** | 0.7764 *** | 0.9326 *** | 0.8852 |
| Financial development level (FINA) | 0.7065 *** | 0.7819 *** | 0.5091 *** | 0.6658 |
| Marketization process (MARK) | 0.9241 *** | 0.7069 *** | 0.8812 *** | 0.8374 |

***, ** and * represent the significance levels of 1%, 5% and 10%, respectively.

Regional technological innovation capability had the strongest positive driving effect on the coordinated development of economic dual circulation, revealing that technological innovation, as a primary productive force, can provide continuous dynamic support for economic development. Its explanatory power reached 0.9653 in 2020 and it became the dominant factor, indicating that with the implementation of the strategy for invigorating China through science and education and high attention to technological innovation ability, the dividend of technological innovation has become increasingly prominent and scientific and technological achievements have increased greatly, which plays a key part in advancing the coordinated development of internal and external circulation. The per capita income level was the core factor of the coordinated development of economic dual circulation. Although the explanatory power slightly weakened, it remained basically stable, manifesting that the regional economic development is the material basis for supporting the healthy development of internal and external circulation and vital for ensuring the coordinated development of economic dual circulation. The level of circulation development had a significant impact on promoting the internal and external circulation development, indicating that the construction and improvement of the circulation system is the basic support. The marketization process and financial development level, which explained the changes in dual circulation coordinated development by 83.74% and 66.58%, respectively, were important factors to realize the coordinated development of internal and external circulation. The smooth flow and rational allocation of elements directly determine the quality of the economic dual circulation. Therefore, a good financial development environment and a high level of marketization will help accelerate the cross-regional and cross-industry flow of elements, optimize element allocation and industrial layout and also promote the quality and coordinated development of dual circulation. The influence of digitization level on the coordinated development of internal and external circulation was positive, the explanatory power increasing from 0.7718 in 2006 to 0.8010 in 2013 and then to 0.8295 in 2020. This showed that with the integrated development of the digital economy and real economy, the continuously released production potential of data elements can bring more new economic models and further expand the development space of economic dual circulation. Although the upgrading of the industrial structure was positively correlated with the coupling coordination degree of economic dual circulation, the effect was not prominent and the intensity was greatly reduced. This reflected that the role of industrial

structure upgrading in promoting the formation of a reasonable regional economic pattern needs to be improved.

From the view of interaction (Figure 6), the interaction between any two factors was stronger than that of single factor. The type of interaction was mainly two-factor enhancement, supplemented by nonlinear enhancement. In particular, in 2006, the interactions between the digitization level and per capita income level, digitization level and circulation development level and financial development level and circulation development level reached 1, reflecting that the digital economy reshaped the social production, circulation and consumption system, accelerated the rapid flow of various resource elements and the deep integration of various market entities and played an important role in realizing the coordinated development of internal and external economic circulation. In 2013, the interaction between the marketization process and financial development level also reached 1, indicating that the two were consistent in promoting the rational allocation of resources and jointly promoted the coordinated development of internal and external circulation. In addition, except for the interaction between per capita income level and industrial economic structure in 2020 being 1, the interaction of the other factors was at a high level, reflecting that the coordinated development of economic dual circulation was the result of multi-factor functions and resonant development.

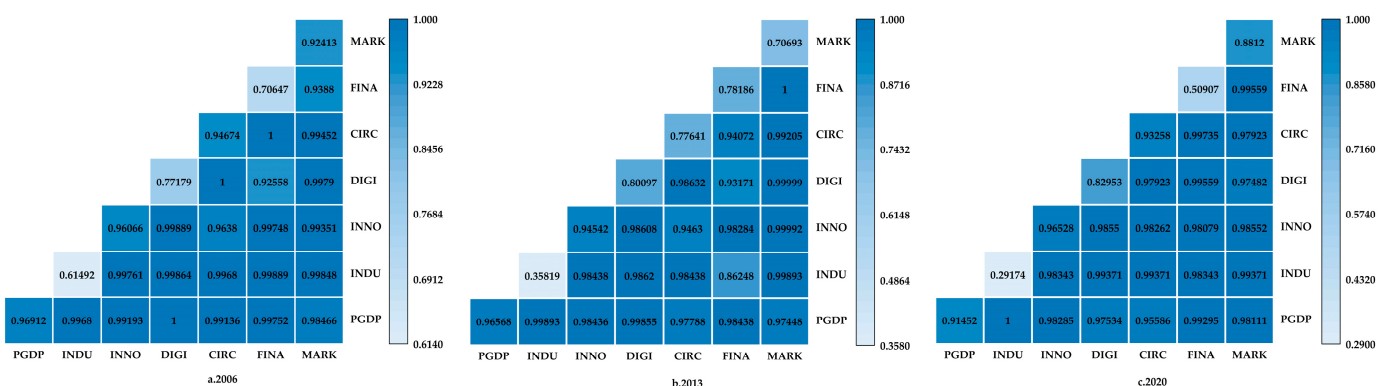

**Figure 6.** Interaction between factors on coupling coordination of economic dual circulation in 2006, 2013 and 2020.

## 5. Discussion

The dual circulation development of the coastal area has a significant effect on the quality and level of high-quality economic development in China [76]. Significant progress has been made in related research, theoretically revealing the relationship between internal and external economic circulation, as well as the driving mechanisms of different factors on the development of economic dual circulation [35,37–42,49–51]. This research further validated the existing research results. Firstly, based on previous research regarding the mutual promotion relationship between internal and external circulation, the commodity circulation, element flow and income distribution were included in the circulation link to construct the evaluation index system of the development level of economic internal and external circulation and the coupling interaction between internal and external circulation was evaluated. The results of this research showed that the coupling coordination relationship between internal and external circulation in the coastal area was significant and had a positive trend. The development quality of internal circulation was better than that of external circulation, and its driving effect on economic development was enhanced, which is consistent with Lv et al. (2022) [77]. Secondly, this paper further analyzed the spatial distribution characteristics of the coupling coordination degree of economic dual circulation in the coastal area from the three dimensions of hierarchical distribution, spatial distribution pattern and spatial agglomeration degree. This paper found that there were obvious differences and agglomeration in the coupling coordination degree of economic dual circulation in the coastal area. The coupling coordination level of economic dual circulation

in Shanghai, Jiangsu and other provinces was higher than that of other coastal provinces, which may be related to the development differences among the provinces themselves. This is similar to the research results of Liu et al. (2022) [51]. Thirdly, this paper also confirmed that a variety of factors could promote the smooth operation and coordinated development of internal and external economic circulation and technological innovation development was the key factor, which is consistent with the results of Wang et al. (2022) and Wang et al. (2022) [22,78]. Meanwhile, the results of this research indicated that the interaction between multiple factors was stronger than the driving effect of a single factor and the coordinated development of economic dual circulation was the result of the comprehensive action of multiple factors.

However, due to data limitations, this paper did not distinguish the internal and external circulation from the perspective of a single province and only took each province as the research unit. Moreover, there are still some driving factors that were not included in the research. Therefore, further exploration should be made in the future from the following aspects: Firstly, the internal and external circulation are relative and each region has its own circulation [79]. Thus, it is necessary to take into account the relativity of internal and external circulation and discuss the performance and interaction relationship of internal and external circulation at a more detailed spatial scale, which will also help to reveal how the coupling coordination relationship between internal and external circulation is influenced by spatial interactions. Secondly, it is necessary to conduct a more in-depth discussion on the testing of driving mechanisms and analyze the driving effects of factors that may be overlooked to enrich the research on the driving factors of economic dual circulation. Thirdly, the research period needs to be extended.

## 6. Conclusions, Implications and Policy Recommendations

### 6.1. Conclusions

This paper used the TOPSIS evaluation model based on the CRITIC-entropy weight method, coupling coordination model, standard deviation ellipse, exploratory spatial data analysis and geographical detector to explore the spatial and temporal evolution characteristics and driving factors of the coordinated development of economic dual circulation in China's coastal area from 2006 to 2020. The main conclusions are as follows:

(1) The changes in the development level of internal and external economic circulation in the coastal area were mainly stable and rising in a fluctuating manner, and the level of internal circulation was better than that of external circulation;

(2) There was significant regional heterogeneity and spatial correlation in the coupling and coordinated development of economic dual circulation in the coastal area, and the degree of spatial dependence showed an inverted U-shaped development trend. The spatial distribution pattern showed the characteristic of "strong in the internal and weak in the external";

(3) The coordinated development of economic dual circulation was the result of multifactor functions and resonant development. Regional technological innovation capability, per capita income level, circulation development level, marketization process, digitization level and financial development level were the core driving forces.

### 6.2. Implications

Theoretically speaking, the four economic links of production, distribution, circulation and consumption in Marx's theory of social reproduction reveal that economic activities are a dynamic circulation process. Based on this and combined with China's experience and the development characteristics of the coastal area, this study introduced the three dimensions of commodity circulation, element flow and income distribution to characterize the circulation link and constructed the evaluation index system for the level of economic internal and external circulation development in China's coastal area, enriching the measurement research of economic circulation. In addition, most scholars stopped at a theoretical analysis of the internal and external circulation relationship and driving

mechanisms. Testing the coordination relationship and the driving factors between the two can serve as a supplement to the relative theories of economic circulation.

In practice, the coastal area is an important node carrying the link between internal circulation and external circulation for China. Analyzing the spatial and temporal characteristics of the coordinated development of its economic dual circulation can provide a good reference for relevant research in other regions and even the whole country. Moreover, this research discussed the driving role in the coordinated development of economic dual circulation from the perspectives of economic development level, circulation system construction and perfection, digital economy integration development and industrial structure upgrading. It was found that the coordinated development of economic dual circulation was not only related to the development level of the region itself but also influenced by multiple factors. This has certain practical significance for achieving the coordinated development of economic dual circulation in different regions.

*6.3. Policy Recommendations*

The coordinated development of economic dual circulation is the key to coping with the complex international situation and achieving high-quality economic development. Based on the research results, the relevant policy recommendations are as follows:

(1) Pay attention to the spatial differences in the coupling coordination degree of economic dual circulation in the coastal area and formulate differentiated strategies to promote the dual circulation development. Northeastern coastal provinces such as Liaoning need to speed up industrial transformation and upgrading and reshape the economic development momentum and vitality;

(2) Make use of the positive spatial correlation of the coupling coordination degree of economic dual circulation, allow full play to the radiation and driving role of provinces with high coupling coordination degrees, such as Jiangsu, Shanghai and Guangdong, and strengthen exchange and cooperation among different provinces, encouraging provinces with low coupling coordination degrees to keep up with provinces with high coupling coordination degrees.

(3) Adhere to innovation-driven development and accelerate key core technology research to enhance the economic system resilience. At the same time, it is necessary to make the most of the joint effect of different driving factors and promote the coordinated development of economic dual circulation.

**Author Contributions:** Conceptualization, J.W. and S.L.; Methodology, J.W.; Data curation, J.W.; Writing—original draft, J.W.; Writing—review & editing, J.W., S.L. and Y.Z. All authors have read and agreed to the published version of the manuscript.

**Funding:** This research was funded by the Key Research Program of the National Social Science Fund of China [grant number 18VSJ06].

**Institutional Review Board Statement:** Not applicable.

**Informed Consent Statement:** Not applicable.

**Data Availability Statement:** The data that were used are confidential.

**Conflicts of Interest:** The authors declare that they have no known competing financial interests or personal relationships that could have appeared to influence the research reported in this study.

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
