# Peer review of "Spatial–Temporal Evolution and Driving Factors of Economic Dual Circulation Coordinated Development in China’s Coastal Provinces"

_sustainability, doi:10.3390/su151411009_

Round 1

Reviewer 1 Report

The article raises an important topic. This paper looked at the coastal provinces in China, and attempted to study the spatial and temporal characteristics and driving factors of the coordinated development of economic dual circulation through an empirical analysis. The research is regional but important for further and wider research. The research is original and the authors put a lot of work into the preparation of the article.

The abstract includes information important for understanding the content of the paper. The methodology is generally clearly defined. Statistical methods were used. Figures and tables are corresponding. The references reflect the topicality of the article. The literature review is good. Results and findings were adequately reported.

Main remarks:

-       The description of the methodology should be supplemented with additional explanations, such as: explain why the hypotheses were not accepted, complete the data source (or is it confidential?) Why other regions of the studied coastline were not included (especially the two in the north)?

-       The conclusions should clearly indicate the contribution of the authors to the literature;

-       It is worth indicating whether the results of the research can be generalized.

Reviewer 2 Report

General Comment

I have now finished reading this manuscript. I believe that the author(s) managed to formulate a clear research question. The topic is a relevant one. I couldn’t find any fundamental flaws in the methodology, data or discussion that would discredit the results.

Some Minor Comments 

Page 3-L-106 please define ADB.

Please give a foot note for the economic meaning of kinetic energy.

* The Kinetic energy (KE) is the Physics to Economics Model makes the analogy that wealth is stored energy. Stored energy in economics is money saved where the money was derived from energy. Electricity plus fuel burned is applied as a force to accelerate the economy. The acceleration of the economy is then demonstrated through the change in the transaction rate over time. In The Physics to Economics Model the energy (E) is equal to one half the economy (e) (as the object to be accelerated) multiplied by the transaction rate squared (E=½e(Tr)²). This means the cause of wealth and the effect (an increase in wealth) are knowable to a degree that enables rational policy to actually increase wealth by intent, which is not the case in current economic thinking (F. P Cunnane, The Physics to Economics Model (PEM, ????))

Reviewer 3 Report

Thank you for this interesting research. This study aims to explore the spatial and temporal evolution of economic dual circulation coordinated development and its driving factors. Hereby are some comments that may help you improve on it:

1.     Please extend the discussions in this paper and link them to the contribution.

2.     Please add the future directions of this research, limitations of the study should also be discussed.

I hope that these notes are helpful in reviewing your article.
